# Development of *lipL32* real-time PCR combined with an internal and extraction control for pathogenic *Leptospira* detection

**Ahmed A. Ahmed** ⊙ *, **Marga G. A. Goris, Marije C. Meijer** ⊙

Department of Medical Microbiology, OIE and National Collaborating Centre for Reference and Research on Leptospirosis, Amsterdam University Medical Center, University of Amsterdam, Amsterdam, the Netherlands

* a.ahmed@amsterdamumc.nl

## Abstract

At least two real-time PCRs for the early diagnosis of leptospirosis have been described, evaluated and validated. However, at least one other report suggested adaptation and modification of primers and probes used in these assays since additional *Leptospira* species have been described and the primers and probe in use possess a serious mismatch to corresponding target sequence. In this study we developed a real-time PCR for detection of pathogenic *Leptospira* based on the *lipL32* gene. The present method consists of generic primers and probes based on target sequence of 10 pathogenic *Leptospira* species including *Leptospira interrogans*. The hybridization, annealing and extension temperature (60°C) were optimized as the optimal temperature of the DNA polymerase enzyme which is used in the amplification reaction. The present assay has a high analytical sensitivity and specificity; the calculated diagnostic sensitivity and specificity were 93.0% and 98.3% respectively. Moreover, the present method includes an internal control which enables easy detection of false negative results and an optional extraction control which enables the estimation of the DNA extraction efficiency.

## Introduction

Real-time PCR in its various forms and chemistries has been applied and adopted in many clinical laboratories as a robust diagnostic tool for detection of pathogens. Particularly, diagnosis of leptospirosis in the early acute phase can facilitate the treatment of an infected patient with a proper antibiotic at an appropriate time, which might prevent further complications including multi-organ failure. In addition, real-time PCR is a perfect tool which can be applied to identify the sources of the infection and related maintenance reservoirs. The diagnosis of leptospirosis in the early acute phase is not possible by serological methods such as ELISA, MAT, and rapid diagnostic tests (RDT's), which mainly rely on detection of anti-leptospiral antibodies and hence only can be detected in late acute phase of the disease (more than 7 days after the onset of the disease) [1, 2]. Detection of DNA of pathogenic leptospires in patient samples is very successful during 1–5 days after the onset of the disease [3]. During the past decade, at least two real-time PCRs have been developed and validated for leptospirosis

**Data Availability Statement:** All relevant data are within the manuscript.

**Funding:** The authors received no specific funding for this work.

**Competing interests:** The authors have declared that no competing interests exist.

diagnosis [3–6]. Since continuously monitoring the performance of these particular molecular tests is required as part of the validation procedure of a diagnostic test [7], at least one other report suggested adaptation and modification of primers and probes used in these assays as many new species have been described and the primers and probes in use possess a serious mismatch to corresponding target sequence [8]. In fact, most of these PCRs targeting pathogenic *Leptospira* have been developed based on the genome sequence of one *Leptospira* species namely *L. interrogans*. Recently, new pathogenic species of the genus *Leptospira* have been described and the whole genome sequence of at least one strain representing each species have been published in the genome sequence database [9], which makes it possible to evaluate *(in silico)* the specificity of the molecular methods used for the detection of this pathogen. According to the OIE recommendation for diagnostic tests; evaluation, monitoring the performance and continuously checking the specificity and the sensitivity of a particular test are critical factors in the validation procedure [7]. In this study, we demonstrate the development and validation of a real-time PCR for detection of pathogenic leptospires according to the OIE criteria. The assay was optimized conforming to the standard real-time PCR protocol using hydrolysis probes (dual-labeled oligonucleotides) [10]. This assay has the ability to detect all pathogenic *Leptospira* species currently known with a high efficiency, sensitivity and specificity. The test has been optimized to accommodate multiplexing with other real-time PCR assays in use for other pathogenic micro-organisms. Alternatively it might be used as a single assay but with another real-time PCR test in the same time using the same thermocycler. Moreover, the assay was optimized including a synthetic template as an internal control (IC) to assure the quality of the system. Optionally, this template could be used to check the efficiency of the DNA extraction procedure as well.

## Materials and methods

The development and validation of this real-time PCR was performed according to the OIE criteria for validation of a diagnostic method [7] and according to a standard real-time PCR protocol using hydrolysis probes [10].

### Ethics statement

This research was exempted from ethical review of human subjects research by the Medical Ethics Review Committee of the Academic Medical Center, University of Amsterdam (W20_327#20.362).

### Micro-organisms and DNAs preparation

In this study, 73 strains belonging to pathogenic, non-pathogenic and intermediate *Leptospira* species (Table 1) and 46 other micro-organisms (Table 2) were included and tested to evaluate the method described in this paper. *Leptospira* strains were derived from the reference collection of the National Leptospirosis Reference Centre (NRL), AMC, Amsterdam, the Netherlands. Genomic DNAs of other 46 micro-organisms were partly acquired from the Microbiology Department (AMC) and partly a gift from colleagues from other institutions.

### DNA extractions

*Leptospira* strains were propagated at 30°C in EMJH liquid media which is prepared according to Ellinghausen and McCullough [11] as modified by Johnson and Harris [12]. The number of bacteria per ml was estimated using a Helber bacteria counting chamber (Weber Scientific international, West Sussex BN15 8TN England). Genomic DNA of *Leptospira* strains from the culture medium and the internal control (IC) template were extracted using Qiagen mini kit

**Table 1. *Leptospira* strains used in the assay.**

| No. | Species | Serovar | Strain | Status | Result | Reference |
|---|---|---|---|---|---|---|
| 1 | *L. alexanderi* | Banna | A 31 | Pathogenic | + | [20] |
| 2 | *L. alexanderi* | Manhao 3 | L 60 | Pathogenic | + | [20] |
| 3 | *L. alexanderi* | Mengla | A85 | Pathogenic | + | [20] |
| 4 | *L. alexanderi* | Manzhuang | A23 | Pathogenic | + | [20] |
| 5 | *L. alstonii* | Pinchang | 80–412 | Pathogenic | + | [20] |
| 6 | *L. borgpetersenii* | Kisuba | Kisuba | Pathogenic | + | [20] |
| 7 | *L. borgpetersenii* | Hardjo typeBovis | Sponselee | Pathogenic | + | [20] |
| 8 | *L. borgpetersenii* | Balcanica | 1627 Burgas | Pathogenic | + | [20] |
| 9 | *L. borgpetersenii* | Mini | Sari | Pathogenic | + | [20] |
| 10 | *L. borgpetersenii* | Kisuba | Kisuba | Pathogenic | + | [20] |
| 11 | *L. borgpetersenii* | Hardjo type Bovis | L550 | Pathogenic | + | [21] |
| 12 | *L. borgpetersenii* | Poi | Poi | Pathogenic | + | [20] |
| 13 | *L. borgpetersenii* | Arborea | Arborea | Pathogenic | + | [20] |
| 14 | *L. borgpetersenii* | Mini | Sari | Pathogenic | + | [20] |
| 15 | *L. borgpetersenii* | Ballum | Mus 127 | Pathogenic | + | [20] |
| 16 | *L. borgpetersenii* | Hamptoni | Hampton | Pathogenic | + | [20] |
| 17 | *L. borgpetersenii* | Kwale | Julu | Pathogenic | + | [20] |
| 18 | *L. borgpetersenii* | Nigeria | Vom | Pathogenic | + | [22] |
| 19 | *L. interrogans* | Copenhageni | Wijnberg | Pathogenic | + | [20] |
| 20 | *L. interrogans* | Pyrogenes | Salinem | Pathogenic | + | [20] |
| 21 | *L. interrogans* | Hardjo type Prajitno | Hardjoprajitno | Pathogenic | + | [20] |
| 22 | *L. interrogans* | Hawain | LT 62–68 | Pathogenic | + | [20] |
| 23 | *L. interrogans* | Waskurin | LT 63–68 | Pathogenic | + | [20] |
| 24 | *L. interrogans* | Copenhageni | M 20 | Pathogenic | + | [20] |
| 25 | *L. interrogans* | Kremastos | Kremastos | Pathogenic | + | [20] |
| 26 | *L. interrogans* | Pomona | Pomona | Pathogenic | + | [20] |
| 27 | *L. interrogans* | Lai | Lai | Pathogenic | + | [20] |
| 28 | *L. interrogans* | Australis | Ballico | Pathogenic | + | [20] |
| 29 | *L. interrogans* | Bataviae | Swart | Pathogenic | + | [23] |
| 30 | *L. interrogans* | Lora | Lora | Pathogenic | + | [20] |
| 31 | *L. interrogans* | Icterohaemorrhagiae | RGA | Pathogenic | + | [20] |
| 32 | *L. interrogans* | Recreo | 380 | Pathogenic | + | [20] |
| 33 | *L. interrogans* | Lai type Langkawi | Langkawi | Pathogenic | + | [24] |
| 34 | *L. kirschneri* | Bim | 1051 | Pathogenic | + | [20] |
| 35 | *L. kirschneri* | Grippotyphosa | Moskva V | Pathogenic | + | [20] |
| 36 | *L. kirschneri* | Cynopteri | 3522 C | Pathogenic | + | [20] |
| 37 | *L. kirschneri* | Grip. type Duyster | Duyster | Pathogenic | + | [20] |
| 38 | *L. kirschneri* | Sokoine | RM1 | Pathogenic | + | [25] |
| 39 | *L. kirschneri* | Lambwe | Lambwe | Pathogenic | + | [20] |
| 40 | *L. kmetyi* | Malaysia | Bejo-Iso[T] | Pathogenic | + | [26] |
| 41 | *L. mayottensis* | Kenya | 200901122 | Pathogenic | + | [27] |
| 42 | *L. meyeri* | Sofia | Sofia 874 | Pathogenic | + | [20] |
| 43 | *L. noguchii* | Huallaga | M7 | Pathogenic | + | [20] |
| 44 | *L. noguchii* | Carimagua | 9160 | Pathogenic | + | [20] |
| 45 | *L. noguchii* | Louisiana | LSU 1945 | Pathogenic | + | [20] |
| 46 | *L. santarosai* | Guaricura | Bov. G | Pathogenic | + | [20] |
| 47 | *L. santarosai* | Luis | M 6 | Pathogenic | + | [20] |

(*Continued*)

**Table 1.** (Continued)

| No. | Species | Serovar | Strain | Status | Result | Reference |
|-----|---------|---------|--------|--------|--------|-----------|
| 48 | *L. santarosai* | Sulzerae | LT 82 | Pathogenic | + | [20] |
| 49 | *L. santarosai* | Rioja | MR 12 | Pathogenic | + | [20] |
| 50 | *L. santarosai* | Huanuco | M 4 | Pathogenic | + | [28] |
| 51 | *L. santarosai* | Varela | 1019 | Pathogenic | + | [20] |
| 52 | *L. weilii* | Ranarum | ICF | Pathogenic | + | [20] |
| 53 | *L. weilii* | Vughia | LT 89–68 | Pathogenic | + | [20] |
| 54 | *L. weilii* | Qingshui | L 105 | Pathogenic | + | [20] |
| 55 | *L. dzianensis* | undesignated | M12A | Pathogenic | + | [29] |
| 56 | *L. barantonii* | undesignated | FH4-C-A1 | Pathogenic | + | [29] |
| 57 | *L. putramalaysiae* | undesignated | SSW20 | Pathogenic | + | [29] |
| 58 | *L. adleri* | undesignated | FH2-B-D1 | Pathogenic | + | [29] |
| 59 | *L. ellisii* | undesignated | AT17-C-A5 | Pathogenic | + | [29] |
| 60 | *L. gomenensis* | undesignated | KG8-B22 | Pathogenic | + | [29] |
| 61 | *L. inadai* | Lyme | 10 | Intermediate | - | [20] |
| 62 | *L. licerasiae* | Varillal | VAR 010 | Intermediate | - | [30] |
| 63 | *L. wolffii* | Khorat | Khorat-H2[T] | Intermediate | - | [31] |
| 64 | *L. broomii* | Hurstbridge type HB6 | 5399 | Intermediate | - | [32] |
| 65 | *L. fainei* | Hurstbridge | BUT 6 | Intermediate | - | [33] |
| 66 | *L. idonii* | Undesignated | Eri-1[T] | Intermediate | - | [34] |
| 67 | *L. biflexa* | Patoc | Patoc I | Saprophytic | - | [20] |
| 68 | *L. biflexa* | Andamana | CH 11 | Saprophytic | - | [20] |
| 69 | *L. meyeri* | Semaranga | Veldrat Sem. 173 Semarang173 | Saprophytic | - | [35] |
| 70 | *L. terpstrae* | Hualin | LT11-33 | Saprophytic | - | [20] |
| 71 | *L. vanthielii* | Holland | Waz Holland | Saprophytic | - | [20] |
| 72 | *L. wolbachii* | Codice | CDC | Saprophytic | - | [20] |
| 73 | *L. yanagawae* | Saopaulo | Sao Paulo | Saprophytic | - | [20] |

[+] Positive PCR result,—Negative PCR result.

(Germany). Genomic DNA of *Leptospira* strains in spiked blood, serum and urine were extracted using easyMAG automated system (bioMérieux, Marcy l'Etoile, France) according to the manufacturer's recommendation. The quality and quantity of leptospiral DNA was estimated by measuring the absorbance of DNA using the spectrophotometer ND-1000 Nanodrop (3411 Silverside Rd, Bancroft Building, Wilmington, DE 19810, USA). *Leptospira interrogans* serovar Icterohaemorrhagiae strain Kantorowicz was used to optimize and evaluate this assay, and therefore a total genome size of 5 Mb was used to estimate the equivalent number of genome copies per µl of the purified leptospiral DNA [13]. All DNAs extracted from *Leptospira* strains (Table 1) were standardized to a concentration equivalent of $10^2$ genome copies/µl. In order to check the efficiency of the extraction method, a concentration of 0.58 pg equivalent to 100 copies of IC DNA was added to the spiked blood, serum, urine and each clinical material prior to DNA isolation. DNA of all patient's diagnostic materials used in this assay such as blood, serum and urine were isolated using the easyMAG automated system.

## Real-time PCR assay

**Primers and probes design and selection.** Primers and probes were designed targeting *lipL32* of pathogenic *Leptospira;* the *lipL32* sequence of 97 strains belonging to 10 species of

**Table 2. Other micro-organisms tested in this study.**

| No. | Species | PCR Result |
|---|---|---|
| 1 | *Acinetobacter calcoaceticus* | - |
| 2 | *Bartonella henselae* | - |
| 3 | *Bacillus subtilis* | - |
| 4 | *Bifidobacterium longum* | - |
| 5 | *Bordetella bronchiceptica* | - |
| 6 | *Borrelia burgdorferi* | - |
| 7 | *Brucella melitensis* | - |
| 8 | *Burkholderia cepacia* | - |
| 9 | *Campylobacter jejuni* | - |
| 10 | *Candida albicans* | - |
| 11 | *Candida dublinensis* | - |
| 12 | *Candida glabrata* | - |
| 13 | *Candida krusei* | - |
| 14 | *Candida parapsilosis* | - |
| 15 | *Corynebacterium diphteriae* | - |
| 16 | *Corynebacterium xerosis* | - |
| 17 | *Enterobacter aerogenes* | - |
| 18 | *Enterococcus faecalis* | - |
| 19 | *Enterococcus faecium* | - |
| 20 | *Escherichia coli* | - |
| 21 | *Helicobacter pylori* | - |
| 22 | *Klebsiella pneumoniae* | - |
| 23 | *Lactobacillus plantarum* | - |
| 24 | *Legionella pneumophila* | - |
| 25 | *Leishmania donovani* | - |
| 26 | *Leptonema illini* | - |
| 27 | *Listeria monocytogenes* | - |
| 28 | *Mycobacterium africanum* | - |
| 29 | *Mycobacterium bovis* | - |
| 30 | *Mycobacterium leprae* | - |
| 31 | *Mycobacterium tuberculosis* | - |
| 32 | *Neisseria gonorrhoeae* | - |
| 33 | *Pasteurella multocida* | - |
| 34 | *Plasmodium falciparum* | - |
| 35 | *Proteus mirabilis* | - |
| 36 | *Pseudomonas aeruginosa* | - |
| 37 | *Rickettsia akari* | - |
| 38 | *Salmonella enterica* | - |
| 39 | *Staphylococcus aureus* | - |
| 40 | *Streptococcus pneumoniae* | - |
| 41 | *Streptococcus sanguis* | - |
| 42 | *Trypanosoma cruzi* | - |
| 43 | *Toxoplasma gondii* | - |
| 44 | *Treponema pallidum* | - |
| 45 | *Turneriella parva* | - |
| 46 | *Yersinia enterocolitica* | - |

- Negative PCR result.

**Table 3. Sequences and modification of the primers, probes and synthetic DNA template IC used to optimize the assay.**

| Oligo ID | Sequence | Target |
|---|---|---|
| LipgrF2 | 5'CGCTGAAATGGGAGTTCGTATGATTTCC3' | *lipL32* |
| LipgrR2 | 5'GGCATTGATTTTTCTTCYGGGGTWGCC3' | *lipL32* |
| LipgrP1 | 5'FAM AGGCGAAATCGGKGARCCAGGCGAYGG3'BHQ1 | *lipL32* |
| IntoF2 | 5'TAGAATCATTGAATCTATCACATCTCATG3' | Internal Control |
| IntoR2 | 5'TTGAACTAAATGTAGACTAAAGATGATCG'3 | Internal Control |
| IntoP1 | 5'TxRd TTCACATTAACATTCAATAATCAATCATGAA3'BHQ2 | Internal Control |
| PlasintS1 | 5'CTATAGAATCATTGAATCTATCACATCTCATGTACTTCACATTA ACATTCAATAATCAATCATGAATTAATTCAATTTCTGATATGAA TCGATCATCTTTAGTCTACATTTAGTTCAATATATC3' | Internal Control DNA template |

pathogenic *Leptospira* were retrieved from the sequence database [14], aligned and a consensus sequence was determined using Mega7 software [15]. Subsequently specific primer sets and probes were designed and modified using free available software (primer3) [16, 17]. The probes were modified at 5' end and 3' end with FAM as a reporter and BHQ1 as a quencher respectively. The sequences of the oligonucleotides, modification of the mismatching base pairs of the primers and probes, and the modification of the 5' end and 3' end of the probes are shown in Table 3.

The sensitivity and specificity of the selected primers and probes were tested *in silico* utilising BLAST (https://blast.ncbi.nlm.nih.gov/Blast.cgi). As a DNA template for the IC, a synthetic DNA sequence was designed, synthesized and cloned in cloning vector pUC57-Kan. A primer set and probe targeting the IC template sequence were designed specifically matching these synthetic DNA sequence. The IC probe was modified at 5' and 3' with TexRed fluorophore and BHQ2 quencher respectively (Table 3). All primers and probes were synthesized by a commercial facility (Sigma-Aldrich, United Kingdom). The IC DNA template was synthesized and cloned by a commercial facility (Thermo Fisher, Germany).

**Optimization of the real-time PCR.** The assay was optimized using LightCycler 480 Probes Master mix (Roche, Germany) according to the manufacturer's recommendation. Two phases of assay optimization were performed. The initial phase was performed without including the IC system. The final optimization phase was performed after the initial one as the IC template, IC primers and probe were incorporated in the reaction mix. In order to achieve optimal performance and maximal PCR efficiency, the selected primers and probe were tested using different concentrations and within a selected range of annealing and hybridization temperatures. The test was performed employing the CFX96 real-time PCR detection system according to the manufacturer's instructions (BioRad Laboratories, Inc, the Netherlands). The selected primers and probe (LipgrF2, LipgrR2 and LipgrP1) sequence are shown in Table 3. In order to get perfect matching and high specificity to all pathogenic *Leptospira* targeted by this assay, the reverse primer LipgrR2 was modified at nucleotide base 18 and 24 with bases Y(C +T) and W (A+T) respectively and the probe LipgrP1 was modified at base 13,16 and 25 with degenerated bases K (G+T), R (A+G) and Y (C+T) respectively. The concentration of the reagents and the cyclic amplification protocol were optimized according to the real-time PCR standard protocol. Briefly, 12.5 µl of 2x master mix (Roche, Germany), 0.4 µM of each leptospires forward and reverse primer (LipgrF2 and LipgrR2), 0.2µM of the leptospires probe (LipgrP1), 0.16 µM of each internal control primers (IntoF2 and IntoR2), 0.08 µM of internal control probe (IntoP2), 0.25 µl double-distilled DNase/RNase-free water and 0.29 pg (equivalent to 50 copies) of IC DNA template, and finally 10 µl of sample DNA template in a total

volume of 25 μl were submitted to the amplification procedure. The amplification program consists of initial DNA denaturation and DNA polymerase activation at 95˚C for 5 minutes, 45 cycles of two steps 95˚C for 20 seconds as denaturation and 60˚C for 30 seconds representing hybridization of the probes, annealing of the primers and the extension of forward and reverse primers.

**Analytical specificity.** The specificity of the assay was investigated using a panel of *Leptospira* strains representing pathogenic, intermediate and saprophytic species (Table 1) and other micro-organisms (Table 2). A concentration equivalent to $10^3$ genome copies per reaction of each *Leptospira* strain was tested in duplicate.

**Analytical sensitivity.** The detection threshold of the PCR was estimated using *L. interrogans* serovar Icterohaemorrhagiae strain Kantorowicz combined with the IC DNA template. Serial dilutions (10-fold) of strain Kantorowicz genomic DNA starting at $1 \times 10^4$ copies per reaction down to 10 copies per reaction were used to construct a standard curve of the assay. To determine the lower concentration which can be detected by the assay, the last positive 10-fold dilution still giving a positive signal was subjected to subsequent 2-fold serial dilutions. Finally the end-point was set at the dilution in which the assay could detect the target in at least 95% of the replicates. As internal control for the clinical samples the concentration of the IC template was standardized to 50 copies per reaction. To assess the effects of the biological matrix on the analytical sensitivity, seronegative (MAT) and PCR negative blood (200 μl) and serum (200 μl) samples as well as PCR negative urine samples (1 ml) were each spiked with a 10-fold serial dilution of 9 x $10^5$ leptospires of *L. interrogans* serovar Icterohaemorrhagiae strain Kantorowic. These samples were subjected to DNA extraction and subsequent amplification. The last positive 10-fold dilution still giving a positive signal was subjected to subsequent 2-fold serial dilutions, DNA extraction and subsequent amplification. PCR data was analyzed using CFX Manager Software (BioRad Laboratories, Inc, the Netherlands) and the regression mode analysis was selected to determine quantification cycle (Cq) values.

**Diagnostic sensitivity (DSe), specificity (DSp) and confidence intervals (CI).** Diagnostic sensitivity, specificity and 95% confidence intervals (DSe, DSp and CI) were calculated using 249 clinical samples of Dutch human patients suspected of leptospirosis. One sample from each patient was tested in this *lipL32* real-time PCR. These clinical samples were submitted to the National Leptospirosis Reference Laboratory (NRL), AMC, the Netherlands. NRL functions as a diagnostic reference center for leptospirosis in the Netherlands and is accredited according to ISO 15189. The standard procedures include MAT, ELISA, culture [2] and *secY* real-timePCR [3] to diagnose leptospirosis. Patients were considered having leptospirosis based on one or more of the following criteria: positive culture, positive PCR, single MAT titer with a pathogenic strain ≥1:160, single IgM-ELISA titer ≥1:160, seroconversion / ≥ four-fold titer rise MAT or IgM ELISA in paired samples taken at least 2 days apart [2].

Serum, EDTA blood and urine samples taken in the period May 2013 until October 2016 were tested. Either banked samples (stored at -20˚C) were used or DNA extracted from these samples (DNA stored < 1 year at -20˚C). All tests were performed in duplicate and the statistical analysis was performed according to standard literature [18, 19].

## Results and discussion

### Real-time PCR profile

*LipL32* primer set LipgrF2/ LipgrR2 and probe LipgrP1 in combination with IC primer set IntoF2/ IntoR2 and probe IntoP1 were selected as they reacted with high sensitivity and specificity to their targets. The optimal concentration of the reaction reagents and the cyclic

amplification conditions were mentioned earlier as "Real-time PCR assay" in the material and methods section.

## Analytical specificity of the assay

*In silico* testing of primer LipgrF2/ LipgrR2 and probe LipgrP1 sequence using BLAST search, shows no identities between the primer set, the probe and genomic sequences of any organism other than pathogenic *Leptospira*. DNA from 73 *Leptospira* strains belonging to 17 pathogenic, 6 intermediate and 6 saprophytic *Leptospira*, as well as 46 other micro-organisms were tested (Tables 1 and 2). Primer set LipgrF2/ LipgrR2 and probe LipgrP1 specifically amplified genomic DNA isolated from pathogenic *Leptospira* species, 1000 genome copies per PCR reaction of all pathogenic strains gave positive results with an average Ct of 31 while they did not react with intermediate and saprophytic species (Table 1) and other micro-organisms (Table 2), indicating a high analytical specificity for pathogenic *Leptospira*.

## Analytical sensitivity of the assay

The analytical sensitivity of the assay when using DNA extracted from *L. interrogans* strain Kantorowicz and IC template was one copy per reaction as the real-time PCR standard curve shows the following values (E = 101,6%, $R^2$ = 0,99, Slope -3,28 and y-int = 41,26) (Fig 1). The regression mode of the analysis was selected in CFX Manager Software for the analysis and determination of the quantification cycle (Cq).

The analytical sensitivity for the spiked serum, blood and urine with *L. interrogans* strain Kantorowicz were estimated as 2, 3 and 5 leptospires per reaction respectively using the protocol mentioned above.

## Efficiency of the extraction method

When using 100 copies of the IC DNA template to estimate the efficiency of the extraction method, we observed 68.3% recovery in serum, 44.7% in blood and 30.0% in urine. Although the estimation of the extraction efficiency is beyond the scope of this study our results show that the IC is useful to evaluate the extraction method.

## Diagnostic sensitivity and specificity

Clinical blood samples from 71 laboratory-confirmed leptospirosis patients and 178 negative patients (suspected of leptospirosis) were enrolled as a prospective consecutive cohort to determine the diagnostic sensitivity (DSe) and specificity (DSp); 66 of the confirmed cases and 3 of the 178 negative cases had a positive *lipL32* real-time PCR result. Therefore the DSe and DSp of the PCR are 93.0% (CI 83.6–97.4%) and 98.3% (CI 94.8–99.6%) respectively.

Sensitivity and specificity of particular real-time PCR assays mainly rely on primers and probe design, their selection criteria and their optimal parameters. Moreover, optimizing the concentrations of all reagents involved in the amplification reaction and optimizing the reaction conditions ensure a sensitive, specific and efficient assay. At least one report illustrated that primers and probes used in two validated PCRs for detection of pathogenic leptospires required modification since serious mismatching in these oligos was reported which resulted in at least one pathogenic species of *Leptospira* which could not be detected [8]. In this assay, the PCR successfully detected all pathogenic *Leptospira* but not saprophytic and intermediate leptospires as well as other microorganism tested so far. This high analytical specificity indicates that the primers and probe set used in this study perfectly matches the target sequences of pathogenic leptospires and excludes saprophytic and intermediate leptospires as well as

other microorganism sequences. Moreover, the oligo set was designed according to a standard procedure and their optimal annealing and hybridization temperature was adjusted to the optimal temperature for DNA polymerase activity (60˚C) permitting a high reaction sensitivity. The assay shows a high analytical sensitivity combined with a high real-time PCR reaction efficiency since one copy of genomic DNA can be detected from pure targeted DNA. However, testing spiked biological materials such as serum, blood and urine with *Leptospira* resulted in an analytical sensitivity of 2, 3 and 5 genomic copies, respectively. The efficiency of the extraction method used to isolate the DNA and the existence of inhibitors may explain the slight differences in the analytical sensitivity value between pure DNA and spiked materials. The diagnostic sensitivity and specificity were calculated as 93.0% and 98.3%, respectively. Calculation of these values mainly rely on the reference standard, in this study diagnostic culturing, serology and *secY* real-time PCR were considered as reference standard according to the standard procedures of NRL.

As quality control, IC was used in this assay to check the performance of the reaction procedure per individual sample and single reaction tube or plate well and to investigate the presence of inhibitors in the clinical samples. Moreover it can be used to check the efficiency of the extraction method. In conclusion, a real-time PCR with high specificity and specificity was developed. This PCR includes the internal control as one of the quality control parameters of this assay and an optional extraction control in case of using a manual and not automated controlled extraction method. The present assay has several advantages over currently in use PCR methods. First it has the capability to detect all known *Leptospira* pathogenic species since the generic primer and probe set used in this assay reacts positively with all mentioned pathogenic species. Secondly the ability to monitor false negative results that may be generated during DNA extraction or during the reaction since the method incorporates the synthetic internal control. Thirdly the hybridization, annealing and amplification temperature (60˚C) have been optimized to the optimal temperature of the DNA polymerase enzyme. This allows multiplexing with other PCRs to detect other pathogens in one tube or to run the test in the same plate but in different wells using the standard cyclic programme.

## Author Contributions

**Conceptualization:** Marga G. A. Goris.

**Methodology:** Ahmed A. Ahmed.

**Validation:** Ahmed A. Ahmed, Marga G. A. Goris, Marije C. Meijer.

**Writing – original draft:** Ahmed A. Ahmed, Marga G. A. Goris.

**Writing – review & editing:** Ahmed A. Ahmed, Marga G. A. Goris, Marije C. Meijer.

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
