## [Decision Letter · Decision Letter 0]

6 May 2020

PONE-D-20-04275

Development of lipL32 real-time PCR combined with an internal and extraction control for pathogenic Leptospira detection

PLOS ONE

Dear Dr. Ahmed,

Thank you for submitting your manuscript to PLOS ONE. After careful consideration, we feel that it has merit but does not fully meet PLOS ONE’s publication criteria as it currently stands. Therefore, we invite you to submit a revised version of the manuscript that addresses the points raised during the review process.

ACADEMIC EDITOR: Statistical evaluation is needed to validate the obtained data.

We would appreciate receiving your revised manuscript by Jun 20 2020 11:59PM. To enhance the reproducibility of your results, we recommend that if applicable you deposit your laboratory protocols in protocols.io, where a protocol can be assigned its own identifier (DOI) such that it can be cited independently in the future. For instructions see: http://journals.plos.org/plosone/s/submission-guidelines#loc-laboratory-protocols

We look forward to receiving your revised manuscript.

Kind regards,

Kalimuthusamy Natarajaseenivasan

Academic Editor

PLOS ONE

Journal Requirements:

"Procedures for collecting patients’ data and use of clinical specimens for laboratory service

improvement falls under the umbrella of the ‘National Coordination Infectious Disease Control’ (Landelijke Coördinatie Infectieziektebestrijding, LCI), ‘Centre for Infectious Disease Control’ (Centrum Infectieziektebestrijding, Cib), which is a formal body of the Netherlands Ministry of Health and resides in the National Institute for Public Health and Environment (RIVM)

in Bilthoven, the Netherlands and thus were conducted in compliance with the regulation, policies and principles of the Dutch Public Health Service Policy. The procedure includes the processing of anonymous data from patients upon receipt of written informed consent."

a.) Please amend your current ethics statement to include the full name of the ethics committee/institutional review board(s) that approved your specific study.

b.) Please amend your current ethics statement to confirm that your named institutional review board or ethics committee specifically approved this study.

Additional Editor Comments (if provided):

Reviewers' comments:

Reviewer's Responses to Questions

**Comments to the Author**

1. Is the manuscript technically sound, and do the data support the conclusions?

Reviewer #1: Yes

Reviewer #2: Yes

2. Has the statistical analysis been performed appropriately and rigorously? 

Reviewer #1: Yes

Reviewer #2: I Don't Know

3. Have the authors made all data underlying the findings in their manuscript fully available?

Reviewer #1: Yes

Reviewer #2: Yes

4. Is the manuscript presented in an intelligible fashion and written in standard English?

Reviewer #1: Yes

Reviewer #2: Yes

5. Review Comments to the Author

Reviewer #1: 1. Page 9, line 127- 128: The basic source of Internal control (IC) is missing. What are the criteria adopted for designing of IC? In which cloning vector used? Is any previous references are available?

2. Page 10, line 136 & 150: Initially it is mentioned that optimization of real time PCR using light cycler 480 probes Master mix (Roche, Germany), later in line 150 2X master mix (Applied Biosystem). Which is exactly used in this study? Kindly modify accordingly

3. Page 11, Line 161 -172: In analytical specificity, 103 genome copies were used. This is equal to 1000 organisms/reaction. Whereas in analytical sensitivity serial dilution (10 fold) of DNA used. Here exact counting of organism is practically difficult. So author may incorporate Initial concentration of DNA/ copy number used for dilution needs to be mentioned, which is missing. Standard curve normalization w.r.t 103 genome copies needs to be incorporated along with the double dilution as well.

4. Page 12, line 175: What is the spiked leptospiral (L.interrogans strain Kantorowic) load used in the first dilution of this study?

5. Page 12, Line 186 -189: Mentioned as “positive PCR. Which type of PCR is missing? Also authors are requested to analyze the data from each case inclusion criteria and draw their positive and negative results obtained from this study. I believe these data may provide certain clues where/in which case(s) the diagnostic efficiency are increased. These findings would helpful for the selection of appropriate sample and its day of collection after onset of disease.

6. Line 105 -106: Mentioned as all leptospiral DNA concentration equivalent to 103 genome/µl. whereas in line 154, 10µl of DNA template was used for the amplification. This is approximately equal to 10,000 copies. In line 163, 103 genome copies per reaction were used. These three statements are having contradicted interpretation and confuse the readers. Authors are requested to modify accordingly.

7. Page 13, line 203-204: any organism other than leptospira; modify other than “pathogenic” leptospira.

8. Line 107 & 172 -173: 100 copies of IC DNA was spiked in clinical specimens for determination of efficiency of extraction. Whether authors are achieved same 100 copies in the extraction? This is missing. Line 172 -173: The concentration of IC template was optimized as 50 copies/reaction. In line 214, considering the above statement, IC template should be two copies/reaction, but it is mentioned one copy reaction. Kindly clarify the statement.

9. Page 14, Line 223 -224: Clinical sample ratios (Positives & Negatives) are statically insignificant. More negative samples were enrolled in the study.

Reviewer #2: The study seems to be a good effort to design generic primers and probes for the diagnosis leptospirosis in the early phase and the manuscript is well written. However, there are few concerns that needs to be addressed. It would be more significant if the statistical methods used to calculate the Diagnostic sensitivity (DSe), specificity (DSp) and confidence intervals (CI) are explained in detail. Moreover, there are few grammatical and spelling errors that needs to be corrected. For instance in Line 17 and 43- ‘target sequence’ has been written as ‘targets sequence’; Line 35 and 227- the word ‘rely’ has been spelt as ‘relay’; Line 89- the word result is spelt twice as ‘PCR result Result”; Line 208- ‘with an average Ct of’ is written as ‘with on average a Ct of’; Line 257- ‘set use in this assay’ should be changed to ‘set used in this assay’; Line 259- change ‘may generated’ to ‘may be generated’, etc

6. PLOS authors have the option to publish the peer review history of their article (what does this mean?). If published, this will include your full peer review and any attached files.

Reviewer #1: Yes: VEDHAGIRI KUMARESAN

Reviewer #2: No

---

## [Author Response · Author response to Decision Letter 0]

21 Jul 2020

1. Page 9, line 127- 128: The basic source of Internal control (IC) is missing. What are the criteria adopted for designing of IC? In which cloning vector used? Is any previous references are available?

Answer: 

This is synthetic DNA, we will change in the text “artificial” to “synthetic”. The criteria are that the sequence of the synthetic DNA does not react with the used target primers sequences and that the primers used to amplify the IC should have an annealing temperature equalling 60 °C as for the target primers. The cloning vector is pUC57-Kan. We have added this information to the manuscript. There is no previous reference available, the IC is designed for the first time described in this manuscript. 

2. Page 10, line 136 & 150: Initially it is mentioned that optimization of real time PCR using light cycler 480 probes Master mix (Roche, Germany), later in line 150 2X master mix (Applied Biosystem). Which is exactly used in this study? Kindly modify accordingly.

Answer: 

Thank you for this remark, we used Roche, Germany for both and we have modified the text accordingly.

3. Page 11, Line 161 -172: In analytical specificity, 103 genome copies were used. This is equal to 1000 organisms/reaction. Whereas in analytical sensitivity serial dilution (10 fold) of DNA used. Here exact counting of organism is practically difficult. So author may incorporate Initial concentration of DNA/ copy number used for dilution needs to be mentioned, which is missing. Standard curve normalization w.r.t 103 genome copies needs to be incorporated along with the double dilution as well. 

Answer:

Initial concentration of DNA has been added to the MS (i.e. strain Kantorowicz genomic DNA starting at 1 × 104 copies per reaction). The standard curve with 10-fold dilution is added to the manuscript as Figure 1. We did not perform a standard curve with doubling dilution, though the limit of detection is mentioned already in the text.

4. Page 12, line 175: What is the spiked leptospiral (L.interrogans strain Kantorowicz) load used in the first dilution of this study?

Answer: 

The load of the strain Kantorowicz species L. interrogans (9x105 copies) of used in the first dilution has been added to the text as well as the volumes of blood, serum and urine.

5. Page 12, Line 186 -189: Mentioned as “positive PCR. Which type of PCR is missing? 

Also authors are requested to analyze the data from each case inclusion criteria and draw their positive and negative results obtained from this study. I believe these data may provide certain clues where/in which case(s) the diagnostic efficiency are increased. These findings would helpful for the selection of appropriate sample and its day of collection after onset of disease. 

Answer 1: 

The PCR is the secY real-time PCR mentioned in Reference 3, we made it more clear in the manuscript.

Answer 2: 

Thank you for your suggestion to analyse the data from each case based on the inclusion criteria, in order to provide possible clues for increased diagnostic efficiency. In our opinion, we might not able to provide these clues. Our case definition is based on the overall results from each patient (serology, culture and secY real-time PCR). We encourage clinicians to submit the appropriate samples according to the days post onset, i.e. 1-10 DPO serum (for serology and PCR), EDTA blood (for PCR) and heparinised blood (for culture), >10 DPO serum (for serology). Urine and CSF can be send at any time point for PCR. However there are limitations to prepare such an analysis as you suggested since the reality is that we often get sub-optimal samples. In this study we also choose to include from each patient only one specimen to perform lipL32 real-time PCR, while the secY real-time PCR is performed on all submitted samples from the patient eligible for PCR (i.e. serum/EDTA blood taken 1-10 DPO, urine and CSF at any timepoint).

In earlier studies it is shown that PCR on blood samples is most useful during the first 10 days of disease (Esteves LM et al. 2018. Diagnosis of Human Leptospirosis in a Clinical Setting: Real-Time PCR High Resolution Melting Analysis for Detection of Leptospira at the Onset of Disease. Sci Rep. 2018;8(1); Ahmed A et al 2009. Development and validation of a real-time PCR for detection of pathogenic leptospira species in clinical materials. PLoS One. 2009;4(9)).

6. Line 105 -106: Mentioned as all leptospiral DNA concentration equivalent to 103 genome/µl. whereas in line 154, 10µl of DNA template was used for the amplification. This is approximately equal to 10,000 copies. In line 163, 103 genome copies per reaction were used. These three statements are having contradicted interpretation and confuse the readers. Authors are requested to modify accordingly.

Answer: 

Thank you for this observation, this is now modified to a concentration equivalent to 102 genome/µl in line 105-106. 

7. Page 13, line 203-204: any organism other than leptospira; modify other than “pathogenic” leptospira.

Answer: 

this is adapted

8. Line 107 & 172 -173: 100 copies of IC DNA was spiked in clinical specimens for determination of efficiency of extraction. Whether authors are achieved same 100 copies in the extraction? This is missing. Line 172 -173: The concentration of IC template was optimized as 50 copies/reaction. In line 214, considering the above statement, IC template should be two copies/reaction, but it is mentioned one copy reaction. Kindly clarify the statement. 

Answer: 

We have added an extra paragraph “Efficiency of the extraction method” showing the results of the recovery efficiency when we use 100 copies of IC. In Line 172-173 we wrote the optimal concentration of IC template was 50 copies/reaction. This is implemented when testing clinical samples as well as spiked material. We have rewritten these lines to make it more clear that there is no relation between the 50 copies/reaction used for the clinical samples and spiked material and the 100 copies/reaction used to estimate the extraction efficiency.

9. Page 14, Line 223 -224: Clinical sample ratios (Positives & Negatives) are statically insignificant. More negative samples were enrolled in the study. 

Answer

Indeed, more negative patients are enrolled in this study. This reflects the pattern of submitted samples to our diagnostic laboratory, so therefore this is a representative population for the Dutch situation.

Points of reviewer#2:

Regarding the statistical method: We have adapted the text, we added the number of cases with a positive lipL32 real-time PCR result for both laboratory confirmed leptospirosis cases and negative cases. 

We feel this is sufficient for the reader to get insight which proportion of the laboratory confirmed leptospirosis cases as well as negative cases have a positive / negative lipL32 real-time PCR result.

We had made already reference to the website < http://vassarstats.net/>, we used as a tool for the statistical computation: Clinical Research Calculators/Calculator 1.

Here the 95% confidence intervals for proportions are calculated according to the efficient-score method (corrected for continuity) described by Robert Newcombe, based on the procedure outlined by E. B. Wilson in 1927. 

We have adapted the grammar and spelling errors according to the remarks.

Points of Academic Editor:

This is addressed above, see reviewer#2, regarding the statistical method.

---

## [Decision Letter · Decision Letter 1]

17 Sep 2020

PONE-D-20-04275R1

Development of lipL32 real-time PCR combined with an internal and extraction control for pathogenic Leptospira detection

PLOS ONE

Dear Dr. Ahmed,

Thank you for submitting your manuscript to PLOS ONE. After careful consideration, we feel that it has merit but does not fully meet PLOS ONE’s publication criteria as it currently stands. Therefore, we invite you to submit a revised version of the manuscript that addresses the points raised during the review process.

We look forward to receiving your revised manuscript.

Kind regards,

Kalimuthusamy Natarajaseenivasan

Academic Editor

PLOS ONE

Reviewers' comments:

Reviewer's Responses to Questions

**Comments to the Author**

1. If the authors have adequately addressed your comments raised in a previous round of review and you feel that this manuscript is now acceptable for publication, you may indicate that here to bypass the “Comments to the Author” section, enter your conflict of interest statement in the “Confidential to Editor” section, and submit your "Accept" recommendation.

Reviewer #1: All comments have been addressed

Reviewer #2: All comments have been addressed

2. Is the manuscript technically sound, and do the data support the conclusions?

Reviewer #1: Yes

Reviewer #2: Yes

3. Has the statistical analysis been performed appropriately and rigorously? 

Reviewer #1: Yes

Reviewer #2: Yes

4. Have the authors made all data underlying the findings in their manuscript fully available?

Reviewer #1: Yes

Reviewer #2: Yes

5. Is the manuscript presented in an intelligible fashion and written in standard English?

Reviewer #1: Yes

Reviewer #2: Yes

6. Review Comments to the Author

Reviewer #1: Dear authors

Authors have addressed all the points raised during preliminary review, however, few incorporation are suggested and requested to include in the final version accordingly.

Reviewer #2: (No Response)

7. PLOS authors have the option to publish the peer review history of their article (what does this mean?). If published, this will include your full peer review and any attached files.

Reviewer #1: **Yes: **VEDHAGIRI KUMARESAN

Reviewer #2: No

---

## [Author Response · Author response to Decision Letter 1]

24 Sep 2020

Dear Dr. Kalimuthusamy Natarajaseenivasan,

Thank you very much for the opportunity to adapt our manuscript ‘Development of lipL32 real-time PCR combined with an internal and extraction control for pathogenic Leptospira detection’ with the last suggestions.

Please find below our response.

Yours sincerely,

A.A. Ahmed PhD

• Page 11; line 170 -171 (Authors are requested to incorporate ): To assess the effects of biological matrix...........sero negative (MAT) and PCR negative blood and serum samples & PCR negative urine samples were spiked with L. interrogans serovar Icterohaemorrhagiae strain Kantorowicz. 

We have incorporated as requested

• Page 14; Line 227 -229: Efficiency of extraction method- Although the spiking was done with 10 fold dilution of 9 X105 leptospires. However, the result mentioned about 100 copies IC only. Kindly clarify/ modify accordingly. 

To clarify: in Material and Methods, page 8 line 100-102 we describe how the efficiency of the extraction method is performed and indeed we used an equivalent of 100 copies of IC DNA and not the 10 fold dilution of 9 X105 leptospires.

• Page14: Line 234: Among 71 positive samples, how many paired samples incorporated in this study from the banked samples? 

There were no paired samples included in the 71 positive samples.

• As mentioned in the line 189, the paired samples should be collected at least 2 days apart. Kindly change the word into 2 weeks. (Refer WHO guidelines). (https://www.who.int/zoonoses/diseases/Leptospirosissurveillance.pdf?ua=1)

Thank you for your remark. Indeed, the WHO guidelines mention to collect paired samples at least 2 weeks apart. However in our laboratory (National Leptospirosis Reference Laboratory (NRL)) we found that a titre rise can even be observed in samples 2 days apart. This is described in the reference [2], already cited in line 185. To clarify, we have cited this reference again at the end of line 189.

---

## [Editor Report · Decision Letter 2]

19 Oct 2020

Development of lipL32 real-time PCR combined with an internal and extraction control for pathogenic Leptospira detection

PONE-D-20-04275R2

Dear Dr. Ahmed,

We’re pleased to inform you that your manuscript has been judged scientifically suitable for publication and will be formally accepted for publication once it meets all outstanding technical requirements.

Kind regards,

Kalimuthusamy Natarajaseenivasan

Academic Editor

PLOS ONE
---

## [Editor Report · Acceptance letter]

22 Oct 2020

PONE-D-20-04275R2 

Development of *lipL32* real-time PCR combined with an internal and extraction control for pathogenic *Leptospira* detection 

Dear Dr. Ahmed:

I'm pleased to inform you that your manuscript has been deemed suitable for publication in PLOS ONE. Congratulations! Your manuscript is now with our production department. 

Kind regards, 

on behalf of

Dr. Kalimuthusamy Natarajaseenivasan 

Academic Editor

PLOS ONE